# Advances in Surgery and Sustainability: The Use of AI Systems and Reusable Devices in Laparoscopic Colorectal Surgery

**DOI:** 10.3390/cancers17050761

**Published:** 2025-02-24

**Authors:** Takuma Iwai, Seiichi Shinji, Takeshi Yamada, Kay Uehara, Akihisa Matsuda, Yasuyuki Yokoyama, Goro Takahashi, Toshimitsu Miyasaka, Takanori Matsui, Hiroshi Yoshida

**Affiliations:** Department of Gastroenterological Surgery, Nippon Medical School, Tokyo 113-8602, Japan; s-shinji@nms.ac.jp (S.S.); y-tak@nms.ac.jp (T.Y.); kay-uehara@nms.ac.jp (K.U.); a-matsu@nms.ac.jp (A.M.); y-yokoyama@nms.ac.jp (Y.Y.); s9057@nms.ac.jp (G.T.); miyasaka@nms.ac.jp (T.M.); t-matsui@nms.ac.jp (T.M.); hiroshiy@nms.ac.jp (H.Y.)

**Keywords:** colorectal cancer, AI, SDG, reusable energy device, laparoscopic-assisted colectomy

## Abstract

This review highlights innovative solutions to address critical challenges in modern surgery, including workforce sustainability, resource recovery, and healthcare cost control. Specifically, it focuses on surgical artificial intelligence (AI) systems that provide real-time visualization of anatomical structures and reusable energy devices that optimize surgical efficiency. These advancements significantly contribute to medical sustainability by enhancing resource efficiency, reducing environmental impact, and maintaining high standards of surgical care, aligning with the Sustainable Development Goals.

## 1. Introduction

The environment in which surgeons currently operate has undergone a significant transformation. The surgical landscape has evolved from traditional open surgery to the widespread adoption of laparoscopic surgery in the 1990s, followed by the introduction of robotic-assisted surgery in the 2000s. The expansion of minimally invasive techniques has enhanced anatomical visualization, particularly of deep-seated organs and delicate membranous structures that were previously difficult to observe with the naked eye. This advancement has not only improved the precision and standardization of laparoscopic surgery but has also influenced open surgical techniques, as surgeons trained in laparoscopy now approach open surgery with a refined anatomical perspective. Furthermore, in the 2020s, the integration of artificial intelligence (AI) into surgical practice is becoming increasingly prominent [1,2]. Additionally, advancements in anesthesiology, perioperative management, and various energy devices [3,4] have led to developments that would have been unimaginable only a few years ago.

Despite these advancements, the sustainability of the surgical workforce remains a global challenge. Issues such as a shortage of surgeons and poor work–life balance in high-demand specialties threaten the continuity of quality care [5,6]. In Japan too, a decrease in the number of surgeons and disparities in medical care between regions are urgent issues, and efforts to regulate overtime work for surgeons have begun [7].

Another pressing issue is the environmental impact of disposable devices that are widely used in surgery. These instruments generate substantial medical wastes, many of which require specialized treatment, contributing to increased greenhouse gas (GHG) emissions and undermining sustainable healthcare [8]. This has resulted in increased GHG emissions, posing a challenge to sustainable healthcare. Current surgical practices that consume disposables incur costs amounting to millions of dollars annually [9], and it is estimated that cost of using reusable devices can be 19 times lesser than that of disposables [10].

Various global issues have been reviewed and collated into a set of 17 specific Sustainable Development Goals (SDGs) that need to be addressed [11]. Surgeons are committed to achieving Goal 3: “Ensure healthy lives and promote well-being for all at all ages”, while also being responsible for providing solutions to mounting challenges, such as healthcare workforce shortages, resource shortages, and rising healthcare costs.

In this review, Section 2 (Chapter I) discusses the overall picture of the challenges facing surgery. Then, as two major measures to address the challenges, Section 3 (Chapter II) introduces the clinical benefits and prospects of surgical support AI systems that visualize the microscopic anatomy of organs in real time, and Section 4 (Chapter III) introduces the current state of the spread of laparoscopic surgery using reusable energy devices and surgical techniques.

## 2. Chapter I: Challenges in Achieving Sustainability in Surgery

In the field of surgery, achieving sustainability requires addressing three major interconnected challenges: (1) securing a sustainable surgical workforce, (2) optimizing resource recovery, and (3) controlling medical costs. These challenges not only represent independent hurdles but also underscore the intricate web of dependencies within modern surgical practice.

### 2.1. Securing the Surgical Workforce

The sustainability of the surgical workforce is crucial for ensuring uninterrupted high-quality care. Factors contributing to the decline in the surgical workforce include the aging of surgeons [12], stress and burnout [13], geographical maldistribution [14], the sub-specialization of surgical specialties [15], and medical students choosing specialties with a better work–life balance [16].

Surgeon burnout, which is exacerbated by long hours and high-stress environments, threatens the continuity of surgical services. Strategic workforce planning and interventions to reduce burnout are imperative [17]. Task shifting is a proven strategy for addressing workforce shortages [18]. Patient outcomes have been proven to be the same whether tasks are performed by trained medical staff or physicians, but this requires appropriate training programs [19].

To ensure sustainability, it is necessary to develop efficient educational frameworks for young surgeons while also enhancing motivation. Additionally, innovative training methods should be established to facilitate task shifting among medical staff, allowing for a more flexible and sustainable surgical workforce.

### 2.2. Optimizing Resource Recovery and Controlling Healthcare Costs

Reliance on disposable surgical instruments and single-use products has a profound environmental impact. Although efforts to recycle single-use devices can be expected to reduce medical waste and costs [20], there are still issues to be resolved in terms of ensuring sterility, maintaining equipment integrity, and complying with legal regulations, and they have not yet been widely adopted worldwide [21,22,23,24]. Transitioning to sustainable practices, such as adopting reusable devices and enhancing waste segregation, is vital [25]. For instance, studies have shown that using reusable instruments can reduce carbon emissions by 40–66% compared with disposable alternatives [26].

At the same time, cost efficiency is another crucial pillar of sustainable surgery. Reducing costs should never mean lowering surgical standards. A reduction in surgical quality could lead to increased postoperative complications, prolonged hospital stays, and higher overall healthcare expenditures. In contrast, laparoscopic surgery for appropriate cases minimizes postoperative pain, shortens hospital stays, and effectively reduces overall medical costs [27]. Similarly, Enhanced Recovery After Surgery (ERAS) protocols lower hospital stays and associated costs while maintaining high-quality perioperative care [28], but this is only possible if optimal surgery is performed.

The key takeaway is that sustainable cost management requires balancing resource recovery, economic efficiency, and surgical quality and safety. Instead of focusing solely on cost cutting, a high-level surgical system should be established that prioritizes resource optimization without compromising patient outcomes.

### 2.3. Interdependency of Challenges

Although these challenges are often discussed as separate domains, they are deeply interlinked. For instance, workforce sustainability impacts resource utilization and cost management. Conversely, cost pressures can lead to workforce strain and limited investment in sustainable practices. These interdependencies highlight the need for holistic solutions.

The following sections explore innovative surgical approaches to address these challenges. We discuss the benefits of incorporating advanced AI systems and reusable energy devices in surgery, which offer promising avenues for aligning surgical practices with sustainability goals.

## 3. Chapter II: Improving Surgical Precision and Educational Support Using AI Technology

Excellent surgery consists of multiple elements, including the collaboration of assistants to expose the surgical field and the surgeon’s appropriate handling and traction of the tissues. Identifying the appropriate layers for dissection is critical [29,30,31]. However, education regarding the appropriate layers is often based on the experience and subjective experiences of skilled surgeons [32,33]. Additionally, concentration can lead to the narrowing of an individual’s field of perception, making it difficult for surgeons to notice the misidentification of the layers during surgery. Therefore, it would be beneficial to enhance the objectivity of the anatomical understanding [34], and AI systems are excellent at providing support in this regard. In Japan, AI systems are being evaluated for their potential to provide objective assessments in the certification process for laparoscopic surgical specialists. These systems analyze key surgical parameters such as dissection plane accuracy, instrument handling consistency, and procedural efficiency, aiming to standardize the assessment of surgical skills [35].

Excluding AI systems dedicated to robotic surgery control, surgical AI in clinical practice can be broadly categorized into two types: (1) intraoperative navigation AI, which provides real-time anatomical mapping and decision support during surgery, and (2) postoperative video analysis AI, which assists in surgical education and performance assessment. Eurekaα (Anaut Inc., Tokyo, Japan) and SurVis-Hys (Jmees, Chiba, Japan) are representative intraoperative navigation AIs, while Theator (Theator Inc., Palo Alto, CA, USA), Touch Surgery Enterprise (Medtronic plc, Dublin, Ireland), and Surgical Vision Eureka (Anaut Inc., Tokyo, Japan) are representative video analysis and education systems. In this paper, we define these two categories collectively as “surgical AI systems” and explore their roles in enhancing surgical precision and education.

Intraoperative navigation AI enables real-time, high-precision mapping of organ anatomy, assisting surgeons by dynamically enhancing visual perception during surgery [36,37,38]. By identifying and highlighting key anatomical structures—such as sparse connective tissue, ureters, nerves, microvessels, and the pancreas (Figure 1)—these systems help distinguish safe dissection planes and alert surgeons to potential anatomical variations. This is expected to improve surgical decision-making and to contribute to reducing the risk of complications such as bleeding and inadvertent organ damage.

Task shifting is an essential initiative for optimizing the sustainability of the surgical workforce. Consequently, the involvement of non-physician medical staff as scopists in laparoscopic surgery is increasing, necessitating the efficient delegation of tasks. The use of AI systems is expected to accelerate the learning curve for anatomical recognition by improving anatomical recognition and providing real-time guidance [34]. In addition to its usefulness in surgery, surgical AI offers advantages in surgical training. Surgical education is a complex process that involves hands-on practice, evaluation, and feedback. AI-assisted postoperative video analysis enables surgical trainees and medical students to objectively compare the dissection layers performed by the surgeon with those recognized by the AI system. This provides a more structured and interactive learning experience beyond merely “watching surgical videos” (Figure 2). By visualizing the correct dissection planes and highlighting anatomical structures in real time, AI enables learners to grasp anatomical variations more effectively. Educational techniques must be updated to keep pace with current technologies. We should consider how to enable the next generation to acquire skills that previously required ten units of effort with only five units of effort. Surgical AI systems could serve as a fundamental pillar in this transformation, fostering a more efficient and standardized approach to skill acquisition. The term “sustainability” may evoke the idea of reverting to pre-modern methods or simply maintaining current practices. However, in the field of surgery, achieving sustainability requires advancing technological integration while optimizing resources and training methodologies. AI-driven surgical support is one symbol of this.

In summary, utilizing AI navigation technology in surgery offers benefits such as (1) enhancing the precision of the surgery itself, (2) facilitating task shifting, and (3) effectively educating trainee surgeons. These effects directly contribute to improving the quality of healthcare, promoting technological innovation, and enhancing system efficiency, thereby supporting SDGs 3: “ensure healthy lives and promote well-being for all at all ages”; 9: “build resilient infrastructure, promote inclusive and sustainable industrialization, and foster innovation”; and 8: “promote sustained, inclusive, and sustainable economic growth, full and productive employment, and decent work for all”, through nurturing young talent and accelerating skill acquisition.

## 4. Chapter III: Sustainable Surgery Through the Use of Environmentally and Economically Considerate Reusable Devices

Following the development and widespread adoption of laparoscopic surgery, various devices such as vessel-sealing systems and ultrasonic-coagulation cutting devices have been developed, significantly improving tissue dissection and hemostasis efficacy. These advancements have played a pivotal role in enhancing and standardizing laparoscopic procedures [39]. However, the problem of spiraling medical costs and the increase in waste and environmental pollution caused by the rapid increase in disposable products have become pressing issues for medical professionals. Addressing these issues requires a sustainable approach that maintains high surgical standards while ensuring economic efficiency and resource optimization.

The most direct solution to these challenges is the use of reusable energy devices, which can significantly reduce surgical waste, optimize hospital resource utilization, and lower overall medical costs. Compared to disposable instruments, reusable devices can undergo multiple sterilization cycles, reducing long-term expenses and minimizing medical waste production. Among reusable energy devices, some models feature a reusable handpiece with replaceable single-use inserts, allowing for partial reuse. Examples of such partially reusable devices include the following: AdTec Bipolar (Aesculap, Tübingen, Germany), MarSeal (KLS Martin, Tübingen, Germany). On the other hand, fully reusable devices that are autoclave compatible and do not require disposable components are currently limited to BiClamp and BiSect (Erbe Elektromedizin GmbH, Tübingen, Germany) [40,41,42,43,44].

While there are reviews on laparoscopic surgery using reusable devices, majority of these studies primarily focus on cost-related discussions [45,46]. However, there is limited research on the practical implementation of reusable instruments within surgical workflows and their impact on clinical outcomes [42,47,48,49]. To establish a truly sustainable surgical system, it is essential to develop strategies for the structured implementation of reusable instruments within standardized surgical workflows. To address the gap in clinical outcome reporting, we present a case study of our institutional experience with reusable energy devices such as BiClamp and BiSect in colorectal surgery, termed Reusable Energy Device Laparoscopic-Assisted Colectomy (RE-LAC). This case study serves as a model to assess the feasibility and safety of reusable energy devices in real-world surgical practice. Colorectal surgery represents an optimal platform for integrating reusable energy devices, as it is widely performed across many facilities worldwide and follows a relatively standardized procedure. Given the high volume of colorectal surgeries conducted globally, the integration of reusable instruments in this field can maximize both economic and environmental benefits.

Reusable energy devices demonstrate hemostatic effects equivalent to those of conventional sealing devices [49,50]. However, there are operational differences that must be considered. Unlike ultrasonic dissection (USD) devices, which integrate both sealing and tissue incision functions, reusable energy devices typically serve a single function, either vascular sealing or tissue incision. This means that hemostasis is performed using a sealing device, while incision is carried out using a bipolar device. As a result, surgeons must develop an efficient method to seamlessly switch between these two instruments. Another key operational aspect is the use of foot pedals, which are necessary due to the compatibility of reusable devices with autoclave sterilization (Figure 3A). Since each device requires a separate pedal, efficient foot control is essential to maintain a smooth workflow and avoid unnecessary movement during surgery. To address these challenges, we propose a “sitting, double-hand method”, in which the foot pedals are placed on the left and right sides, allowing the surgeon to operate the sealing device with the left hand and the bipolar device with the right hand. This setup minimizes unnecessary body movement during foot pedal operation and prevents time loss caused by switching between devices (Figure 3B). By adopting this approach, surgeons can eliminate unnecessary time loss associated with foot pedal switching and improve overall efficiency during surgery. Further details and the key points of this method are explained in another report [51]. Additionally, we recommend the use of a dedicated foot pedal mat to improve stability and ensure consistent pedal control (Figure 4). In summary, while reusable energy devices require a slight adjustment in surgical technique, implementing these optimizations can greatly enhance their usability and efficiency in laparoscopic procedures.

As preliminary reference data of RE-LAC, we present the results of a retrospective analysis of laparoscopic-assisted colectomy (LAC) for left-sided colorectal cancer (sigmoid colon, rectosigmoid, and upper rectum) performed at our institution between April 2023 and December 2024. The patients were categorized into two groups: The RE-LAC group (*n* = 17) and the SUD-LAC group (*n* = 35), in which conventional single-use devices (SUDs) were employed (Table 1). All the procedures were performed by Japan Society for Endoscopic Surgery-certified surgeons. The RE-LAC group included 17 cases (7 sigmoid colon, 6 rectosigmoid, and 4 upper rectum), and the SUD-LAC group included 35 cases (11 sigmoid colon, 15 rectosigmoid, and 9 upper rectum). The median operation time was 190 min for RE-LAC and 176.5 for SUD-LAC, and the median blood loss was 0 mL for RE-LAC and 0 mL for SUD-LAC, indicating no significant difference between the two approaches (operation time, *p* = 0.18; blood loss, *p* = 0.66) (Figure 5). These findings serve as preliminary reference data on the safety and feasibility of RE-LAC while also laying the groundwork for further investigation in this field. Notably, previous studies in this field, such as those by Yoshioka [42] and Hasanov [49], have also been based on relatively small cohorts, suggesting that there is still much room for research on reusable energy devices. Future large-scale, multi-center cohort studies are essential to refine the case selection criteria for the RE device and the SUD and to validate their long-term clinical impact.

RE-LAC enables safe and effective laparoscopic colectomy by enhancing the understanding of device operation and optimizing resource utilization. This technique offers both economic and environmental benefits, making it a viable option for sustainable colorectal surgery. By maintaining the quality of surgeries while reducing medical costs and environmental impacts as well as conserving energy related to the disposal of waste, RE-LAC contributes to the SDGs 3: “Ensure healthy lives and promote well-being for all at all ages”; 12: “Ensure sustainable consumption and production patterns”; and 13: “Take urgent action to combat climate change and its impacts”.

We summarize the contents of Section 3 (ChapterII) and Section 4 (ChapterIII) and outline the contribution of the surgical AI system and RE-LAC to the SDGs in Figure 6.

## 5. Discussion

The surgical landscape is constantly changing; therefore, we must adopt a flexible attitude that actively incorporates the latest technologies and devices available rather than continuing with traditional methods [52]. Improving the surgical work environment will lead to an increase in the number of young surgeons and an improvement in the level of surgical treatment, ultimately benefiting patients undergoing surgery [53,54]. We discuss the issues and prospects for the two themes discussed in this paper, surgical AI systems and reusable energy devices, and consider further efforts to contribute to the SDGs in surgery.

### 5.1. Expanding Applications of Surgical AI

A key advantage of surgical AI systems is its extremely simple method, which only requires connecting the SDI cable to existing laparoscopic towers, and its display of structures, which is pixel-based rather than flat and is presented naturally according to confidence levels. Additionally, it continues to learn from many surgical videos, accumulates data, and is updated every few months to improve analysis accuracy. Currently, as a system available for actual surgeries, “Eureka α” has received medical device approval for connective tissue recognition in gastric, colorectal, and hernia surgeries. Future approvals for functions such as nerve and pancreatic recognition are planned. If real-time display without delay becomes possible, it will allow for actual monitoring or surgeries conducted alongside monitoring, significantly contributing to the reduction of surgical complications and the alleviation of technical disparities. To further improve the analytical capabilities of surgical AI systems, it is necessary to continue collecting unbiased, diverse data and to follow strict verification protocols. Currently, surgical AI systems analyze exposed structures; however, if it is possible to predict and detect structures covered by fatty tissue in the future, it will be possible to create a system that can avoid problems in cases such as abnormal blood vessel courses. Such a system would also be very effective in the fields of respiratory and neurosurgery, as well as in the abdominal region.

Traditionally, surgical education has been a type of apprenticeship, with instructions given from senior to junior [33]. By making good use of the visualizations and educational benefits of microscopic dissection of surgical AI systems, it is possible to expect the same benefits as those taught by a skilled surgeon. This may be particularly beneficial in small-scale facilities in which the average age of surgeons is high and there are few surgeons, as well as in rural facilities. In addition, if medical students experience this type of education, there is a possibility that more young people will become interested in becoming surgeons in the future.

When utilizing AI support in surgery, it is important to remember that the foundation of surgery is the skills, experience, and judgment of the surgeons themselves. AI can only demonstrate its full potential when it operates synergistically with a highly trained and skilled surgeon [55]. Therefore, fostering the development and refinement of surgical expertise through rigorous training and continuous education is essential [56]. The integration of surgical AI must not overshadow the critical importance of honing the craft of surgery itself.

### 5.2. Expanding Applications of Reusable Energy Devices

RE-LAC involves two types of devices, and the double-foot switch complicates the operations. Thus, it is recommended to start with simpler procedures, such as sigmoid colon resection. Once familiar with the surgery, surgeons can perform rectal and transverse colon surgeries. In the future, it is hoped that the applications of reusable energy devices will expand to other gastrointestinal cancers and laparoscopic surgeries in other fields, leading to a shift in the awareness of resource recovery and healthcare economics across the entire surgical field.

The pedal mat mentioned earlier was designed to be placed at the same width as the Da Vinci foot switch (Figure 4B*). Additionally, the use of a scope holder provides a stable surgical field. Thus, the combination of a double-foot switch, seated position, and stable visual field allows for surgeries that anticipate transitions to and training for robot-assisted surgery, which is another significant feature of RE-LAC [51].

Reports on laparoscopic surgeries using reusable energy devices have been published in various fields, including thoracic, hepatobiliary, pancreatic, and gynecologic surgery [40,42,43,48,49,57,58,59]. To promote the widespread adoption of reusable devices, interdisciplinary collaborations should be encouraged, such as forming working groups dedicated to surgical sustainability. At the same time, it is crucial to maintain a flexible approach to surgical device selection. While the advantages of reusable devices are evident, not all cases should mandate their use. The decision to use reusable or disposable instruments should be made based on the specific surgical technique and the surgeon’s proficiency. This balanced decision-making can help maintain surgical quality while maximizing resource efficiency. To achieve this, large-scale studies are needed to establish clear selection criteria for cases in which reusable or disposable instruments are most appropriate.

For surgical systems that are beneficial for resource recovery and cost to become more widespread in the future, it is hoped that the development of other reusable energy devices and the recycling of disposable energy devices will be enhanced and widely disseminated. Reusable devices, however, face hurdles related to sterilization, regulatory compliance, and the initial investment required for adoption. Collaborative efforts among healthcare providers, manufacturers, and policymakers are essential to streamline these processes and reduce barriers to entry.

### 5.3. Global Relevance and Accessibility

AI systems and reusable energy devices together are transformative tools for achieving sustainable and equitable surgical care worldwide. Their integration addresses critical challenges such as workforce shortages, resource constraints, and environmental impacts, especially in low- and middle-income countries (LMICs) [60].

In LMICs, where access to experienced surgeons and advanced equipment is limited, surgical AI systems can bridge skill gaps by providing real-time anatomical guidance and decision support. Reusable devices offer a cost-effective solution by reducing dependence on single-use instruments and lowering both procurement and disposal costs. The environmental benefits of adopting reusable devices are significant, with potential reductions of up to 66% in medical waste and greenhouse gas emissions. Similarly, economic advantages, such as reduced procurement and waste management costs, make these devices an attractive choice even for resource-constrained facilities. AI further supports these efforts by improving surgical efficiency and minimizing complications, thereby reducing the overall healthcare costs.

The successful implementation of these technologies depends on policy support and international collaboration. Governments can incentivize adoption by subsidizing initial investments in reusable devices and AI systems while establishing clear guidelines to ensure their safe and effective use. Collaborative efforts among stakeholders can also drive innovation and facilitate the development of affordable, scalable solutions tailored to LMICs.

## 6. Conclusions

This review outlines initiatives using AI systems and reusable energy devices to address the challenges faced in surgery, such as the sustainability of the surgical workforce and environmental considerations worldwide. Through efforts toward sustainable surgical healthcare, these technologies have great sustainability value for healthcare professionals, patients, and the environment. Several steps are recommended to fully realize the potential of these innovations (Figure 7).

Expansion of AI capabilities: Developing algorithms capable of recognizing a wider range of anatomical structures and surgical scenarios will enhance the utility of AI systems across different disciplines.Enhancement of reusable device infrastructure: Investing in research and development for advanced sterilization methods and durable materials will facilitate the broader adoption of reusable devices.Policy and collaboration: Policymakers should establish incentives for hospitals to adopt sustainable practices, including subsidies for initial investments in reusable technologies and AI systems.Education and training: Incorporating AI-assisted training modules into surgical education curricula will ensure that the next generation of surgeons are well versed in sustainable practices.

Regardless of how wonderful a system is, if the hurdles to its introduction are high, it will be of little significance. The important points about the surgical AI and RE-LAC systems, which we have focused on in this article, are that they are intuitive and easy to understand and that they can be easily introduced into any facility. We hope that these initiatives for sustainable surgical practices will spread to other medical fields, thereby prompting a shift in awareness and improving the overall sustainability of healthcare systems.

## Figures and Tables

**Figure 1 cancers-17-00761-f001:**
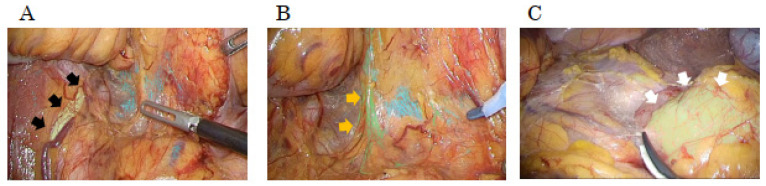
Real-time visualization of anatomical structures using the surgical AI system. The surgical AI systems enable the identification of important anatomical structures, provide visual feedback through coloring, and clearly display the layers. Examples: (**A**) ureter (black arrow), (**B**) nerve (orange arrow), and (**C**) pancreas (white arrow).

**Figure 2 cancers-17-00761-f002:**
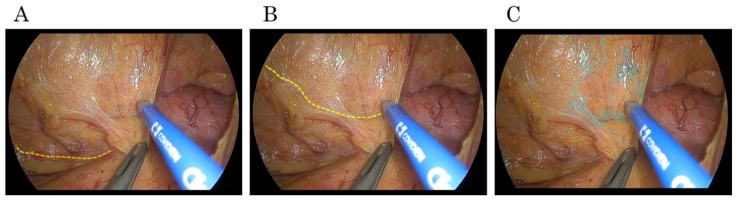
Educational advantages of surgical AI systems. (**A**) The incision layer created by a surgical trainee (orange line). (**B**) Layer in which the retroperitoneal tissue is properly dropped onto the dorsal side by a board-certified laparoscopic surgeon (yellow line). (**C**) Layer of connective tissue as shown by surgical AI systems in postoperative analysis. It can be observed that this is almost the same as the dissection layer performed by an experienced surgeon.

**Figure 3 cancers-17-00761-f003:**
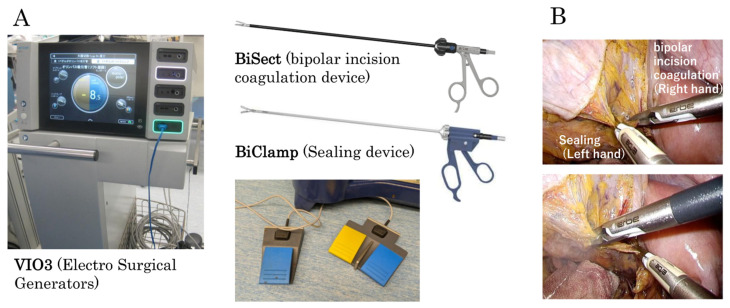
RE-LAC “double-hands method”. Reusable energy devices and their respective foot pedals (**A**). BiSect and BiClamp play different roles. By operating the sealing device with the left hand and the bipolar device with the right hand, the time loss caused by changing between the devices was eliminated (**B**).

**Figure 4 cancers-17-00761-f004:**
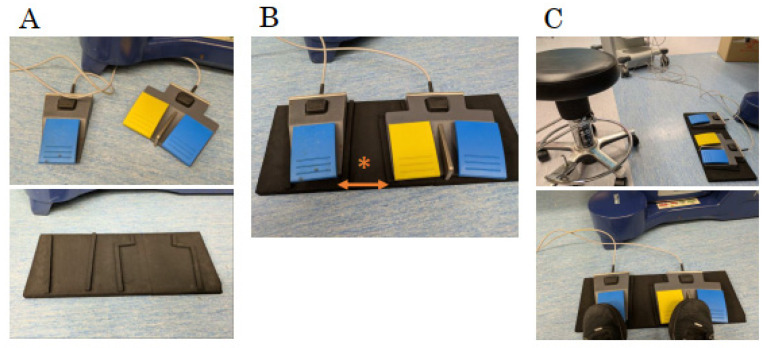
Sitting position and foot pedal setup for the RE-LAC procedures. We created a special mat on which to place the foot switch to improve stability (**A**). We designed the spacing to be the same width as the Da Vinci footswitch * (**B**). Surgeons can sit down and stabilize the foot-switch operation with both feet (**C**).

**Figure 5 cancers-17-00761-f005:**
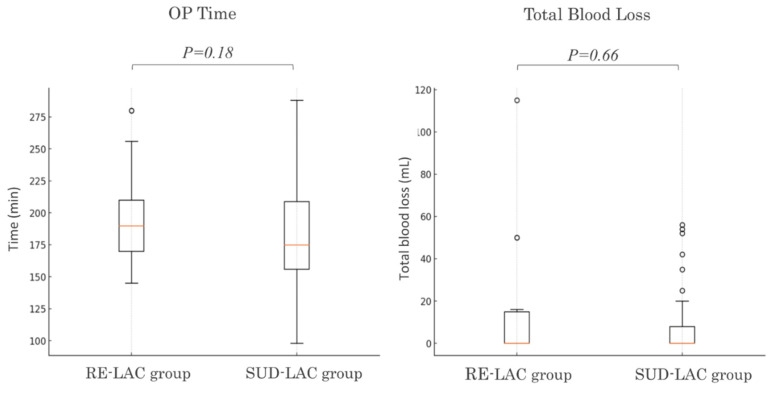
Comparison of clinical outcomes: RE-LAC vs. SUD-LAC. Comparison of the operative times and blood loss volumes of laparoscopic colorectal surgery between the RE-LAC and SUD-LAC groups No significant differences were observed between the two groups.

**Figure 6 cancers-17-00761-f006:**
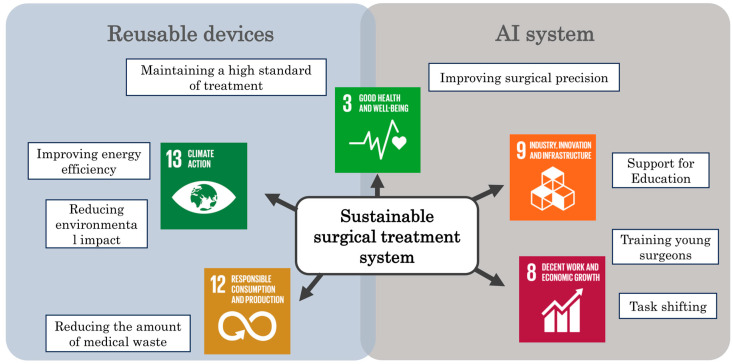
Contribution of reusable devices and AI systems to achieving the SDGs. This diagram shows how AI and reusable surgical devices can provide surgical support to contribute to achieving a sustainable surgical treatment system.

**Figure 7 cancers-17-00761-f007:**
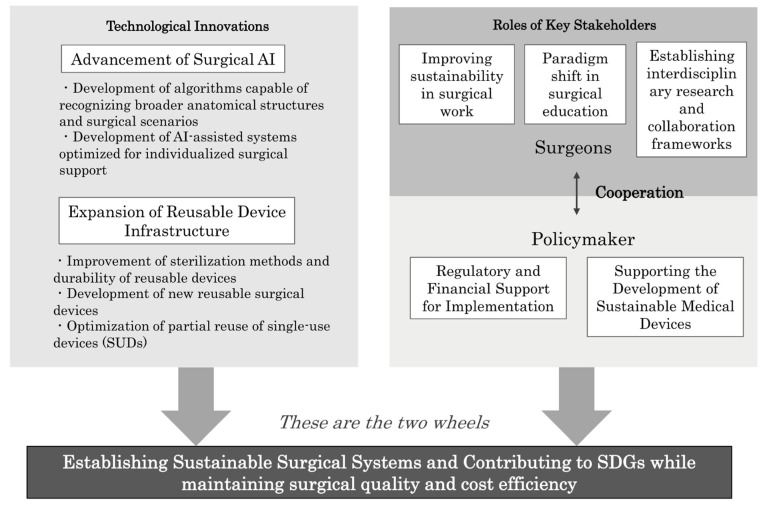
Roadmap for sustainable surgical practices. This figure outlines key strategies for achieving surgical sustainability through AI systems and reusable devices. It highlights advancements in AI-assisted surgery, improved infrastructure for reusable instruments, policy support for adoption, and AI-integrated training to enhance surgical education and efficiency.

**Table 1 cancers-17-00761-t001:** Patient characteristics.

		Total	RE-LAC Group	SUD-LAC Group
		*n* = 52	*n* = 17	*n* = 35
Patient	Age (y/0)	66 (57.5–76.25)	67 (61.0–79)	69 (55–74)
	Sex [Male:Female]	29:23	8:9	21:14
Primary tumor	pT [T1b:T2:T3:T4a]	2:15:25:9	0:5:9:3	2:10:16:6
	pN [N0:N1:N2:N3]	31:18:3:0	9:7:1:0	22:11:2:0
	Neoadjuvant chemotherapy before colectomy [yes] (%)	13.40%	11.7%	14.30%
OP	Operative time (min)	189 (163–210)	190 (179–209)	176.5 (156–209)
	Total blood loss (mL)	0 (0–10)	0 (0–14.5)	0 (0–8.75)
		median (interquartile range)	median (interquartile range)	median (interquartile range)

## Data Availability

The datasets generated and analyzed in the current study are available from the corresponding author upon reasonable request.

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
