# Peer review of "Advances in Surgery and Sustainability: The Use of AI Systems and Reusable Devices in Laparoscopic Colorectal Surgery"

_cancers, 2025, doi:10.3390/cancers17050761_

Round 1
Reviewer 1 Report
Comments and Suggestions for Authors
Dear Author,
Thanks for an excellent article "Advances in surgery and sustainability: the use of AI systems 2 and reusable devices in laparoscopic colorectal surgery" with a lot of information and future of the surgery. It is having strength of AI and Biodevices use with justification and proper disposal. It is a good article which may be improved if following points are considered
1. Data, evidence and literature should be added which is a little less in this manuscript.
2. Number of cases mentioned in this manuscript is just 37 which is a small figure.
3. EUREKA and Bilac system can be an adjunct in surgical practice which will be explored more in future.
4. In summary, a flow chart should be added.
Thanks
Author Response
Dear. Reviewer 1
Thanks for an excellent article "Advances in surgery and sustainability: the use of AI systems 2 and reusable devices in laparoscopic colorectal surgery" with a lot of information and future of the surgery. It is having strength of AI and Biodevices use with justification and proper disposal. It is a good article which may be improved if following points are considered
>>Thank you for your kind words and for recognizing the significance of our work. We greatly appreciate your constructive feedback, which has helped us refine and strengthen our manuscript.
- Data, evidence and literature should be added which is a little less in this manuscript.
>>Thank you for your valuable feedback. Based on your suggestion, we have restructured some sections of the manuscript and added more supporting references to strengthen the scientific foundation of our discussion. Specifically, we have incorporated additional literature on surgical AI systems, the clinical outcomes of reusable energy devices, and the environmental impact of disposable versus reusable instruments. We appreciate your recommendation, which has enhanced the comprehensiveness of our review.
- Number of cases mentioned in this manuscript is just 37 which is a small figure.
>>Thank you for your insightful comment. We acknowledge that our study cohort is relatively small. However, while cost comparisons of reusable devices have been reported in multiple studies, clinical outcome data specifically on reusable energy devices remain scarce. For example, Yoshioka et al.'s study on liver resection using reusable devices also involved a small cohort (n=50). The fact that large-scale studies have not yet been conducted suggests that this is an emerging field with significant potential for future development.
That being said, we fully recognize the limitations of a small sample size, and thus, our data should be interpreted as preliminary, providing reference data rather than definitive conclusions. However, given that laparoscopic colorectal surgery is widely performed worldwide, our study offers one of the first structured implementations of reusable devices in this high-volume surgical field, making it a valuable dataset.
In response to your suggestion, we have extended the observation period to December 2024, increasing the sample size to Bi-LAC n=17 vs. SUD n=35 (total n=52). The statistical significance remains unchanged, but this additional data strengthens the reliability of our findings. We believe this study lays the groundwork for future large-scale, multi-center investigations that will be necessary to further evaluate the safety and efficacy of reusable energy devices in surgery.
- EUREKA and Bilac system can be an adjunct in surgical practice which will be explored more in future.
>> We completely agree with your perspective. We hope that this paper serves as a foundation for active discussions on how these systems can be widely adopted and optimally utilized by surgeons in clinical practice.
Additionally, to reduce the frequent use of product names, we have made terminological adjustments:
"Eureka" has been referred to as "surgical AI."
"Bi-LAC" has been renamed "RE-LAC" (Reusable Energy Device Laparoscopic-Assisted Colectomy).
These changes enhance clarity and generalizability, allowing for broader discussions on the integration of AI and reusable devices in surgical practice.
- In summary, a flow chart should be added.
>> "Thank you for your valuable suggestion. We have visualized the future outlook using a flow chart (Figure 7), which illustrates the cooperative role of technological innovation and policy in achieving sustainable surgical systems."
This makes it clear why the chart was added and how it contributes to the manuscript.
Reviewer 2 Report
Comments and Suggestions for Authors
Advances in surgery and sustainability: the use of AI systems and reusable devices in laparoscopic colorectal surgery
Both topics are of utmost importance and interest.
The authors deliver the necessary general information on issues such as a shortage of surgeons, the environmental impact and cost of disposable devices that are widely used in surgery.
The authors remind that education regarding the dissecting technique is often based on the experience of skilled surgeons
The Surgical Vision Eureka system allows precise mapping of organ anatomy in real-time during surgery, providing surgeons with numerous fields of view and offers connective tissue recognition in gastric, colorectal, and hernia surgeries. The perspectives are to predict and detect structures covered by fatty tissue in the future.
However, the problem of spiraling medical costs and the increase in waste and environmental pollution caused by the rapid increase in disposable products have become pressing issues for medical professionals, and there is a need to maintain high surgical standards while also considering economic efficiency and resource recovery. In this regard the authors present their comparative experience using reusable BiSect and BiClamp with those of laparoscope-assisted colectomy performed using conventional single-use devices. The duration is comparable with negligible blood loss in both.
To improve the stability of the foot pedals while operating the authors recommend creating a dedicated mat, illustrated with pictures.
For a wider penetration of reusable instruments there are necessary initiatives that transcend departments, such as holding working groups on surgery and using reusable equipment. Hurdles are identified - sterilization, regulatory compliance, and the initial investment required for adoption.
The information covers the topics, which are in fashion.
The review is clear, comprehensive and of relevance to the field and has the merit of containing basic information available at the moment but also brings its own arguments in favor of reusable devices.
I found a much more detailed paper published after this one was submitted: Eussen MMM, Moossdorff M, Wellens LM, et al. Beyond single-use: a systematic review of environmental, economic, and clinical impacts of endoscopic surgical instrumentation. Int J Surg. 2024;110(12):8136-8150. Published 2024 Dec 1. doi:10.1097/JS9.0000000000002141.
The references are appropriate, up-to-date and contain over 52 titles. I found 2 self-citations which are logical and appropriate showing the interests of the authors.
The figures (6) are of good quality, appropriate and original. They illustrate and help understanding the text.
The table offers the figures for the comparison between reusable and single use devices.
The conclusions and future are detailed, coherent and connected to the content.
In my opinion the paper fits the journal and the language is correct and understandable.
I recommend the paper to be accepted.
Author Response
Dear Reviewer2,
Thank you for your positive evaluation and recommendation for acceptance. We greatly appreciate your thoughtful review and acknowledgment of the significance of this study in addressing surgical workforce sustainability, economic efficiency, and environmental concerns in modern surgery.
We are particularly grateful for your recognition of the clarity, relevance, and originality of our work, as well as your positive assessment of our figures, tables, and references.
Regarding the additional reference you provided (Eussen et al., 2024), we appreciate this valuable suggestion. We have reviewed the paper and incorporated relevant insights into our discussion where appropriate, further strengthening the manuscript’s foundation.
Once again, we sincerely appreciate your constructive feedback and recommendation for acceptance.
Reviewer 3 Report
Comments and Suggestions for Authors
I read with interest the manuscript by Iwai et al. titled ‘Advances in surgery and sustainability: the use of AI systems and reusable devices in laparoscopic colorectal surgery’. The manuscript covers the important topic of sustainability with interesting insights on AI and devices in colorectal surgery.
Please see my comments presented point-by-point:
· It may be redundant to list the chapters and their content in the introduction. To improve readability, I suggest rephrasing this section to generally refer to the content of the chapters without explicitly listing them.
· Point 3 seems to fall between two chapters and appears shorter compared to the other sections. Would it be better to rename it as 2.1? There may not be any advantage in repeating the content of the chapters in this section.
· Considering the nature of the topic and the utmost importance of the various themes, I find the explicit mention of commercial devices to be inappropriate. The authors present a study that could be included in an interesting and much-needed review.
· The authors do not mention any of the mandatory steps for a clinical trial involving human subjects, yet they report a cohort of cancer patients with operative outcomes.
· I suggest the authors limit their research to the topics outlined in the title and focus on sections that do not refer to commercial devices. The topic has potential and is of growing interest in the surgical community.
Author Response
Dear Reviewer 3,
I read with interest the manuscript by Iwai et al. titled ‘Advances in surgery and sustainability: the use of AI systems and reusable devices in laparoscopic colorectal surgery’. The manuscript covers the important topic of sustainability with interesting insights on AI and devices in colorectal surgery.
>>Thank you for your thoughtful and constructive review of our manuscript. We appreciate your valuable insights and suggestions, which have helped us improve the clarity, focus, and rigor of our work.
It may be redundant to list the chapters and their content in the introduction. To improve readability, I suggest rephrasing this section to generally refer to the content of the chapters without explicitly listing them.
>> Thank you for your insightful suggestion. As you pointed out, some parts of the introduction were repetitive and unnecessarily lengthy. To improve readability, we have revised and streamlined the introduction by removing the explicit listing of chapter content while maintaining a clear overview of the manuscript’s structure. This modification enhances the flow of the introduction and avoids redundancy.
Point 3 seems to fall between two chapters and appears shorter compared to the other sections. Would it be better to rename it as 2.1? There may not be any advantage in repeating the content of the chapters in this section.
>> Thank you for your valuable feedback. As you correctly pointed out, there was some redundancy between point (2) and (3), which affected the manuscript’s structural clarity. To address this, we have merged these sections into a single, more comprehensive point, ensuring a clearer and more logical flow. This revision eliminates unnecessary repetition while maintaining the integrity of our arguments.
Considering the nature of the topic and the utmost importance of the various themes, I find the explicit mention of commercial devices to be inappropriate.
>>Thank you for your thoughtful feedback. We agree that focusing on specific commercial products could limit the generalizability of our discussion. In response to your suggestion, we have minimized direct references to product names and instead emphasized broader concepts such as AI-assisted surgical systems and reusable energy devices.
Additionally, we have revised the chapter titles to ensure consistency with this approach. Furthermore, to maintain a more neutral and conceptual framework, we have changed the abbreviation "Bi-LAC" to "RE-LAC" (Reusable Energy Device Laparoscopic-Assisted Colectomy) to better reflect the study’s focus on reusable surgical technology rather than specific brands.
This adjustment allows for a more objective discussion on surgical sustainability while maintaining the clarity and impact of our analysis.
The authors present a study that could be included in an interesting and much-needed review.
>>Thank you for your encouraging feedback. While we acknowledge that our study is based on a small cohort, we believe it provides valuable clinical data on the systematic use of reusable energy devices in colorectal surgery. Given the scarcity of clinical outcome reports on reusable energy devices, we consider this study to be an important contribution to the field.
This area of research holds significant potential for future development, and large-scale, multi-center studies will be necessary to further validate the safety and efficacy of reusable surgical systems. We sincerely hope that our work serves as a foundation for future investigations and helps stimulate further research in this promising field.
The authors do not mention any of the mandatory steps for a clinical trial involving human subjects, yet they report a cohort of cancer patients with operative outcomes.
>> Thank you for pointing this out. We have now included additional information regarding ethical approval and institutional review board (IRB) approval in the manuscript to ensure clarity on the ethical considerations for our study.
All procedures involving human subjects were conducted in accordance with ethical guidelines, and approval from the relevant ethics committee was obtained. Details on the approval process, patient consent, and compliance with research ethics regulations have been explicitly stated in the revised manuscript.
We appreciate your feedback, which has helped improve the transparency and rigor of our study presentation.
I suggest the authors limit their research to the topics outlined in the title and focus on sections that do not refer to commercial devices. The topic has potential and is of growing interest in the surgical community.
>> Thank you for your valuable suggestion. In response to your feedback, we have revised the manuscript to avoid focusing on specific commercial products and instead framed the discussion around general concepts, such as AI-assisted surgical systems and reusable energy devices.
This revision ensures that the study remains aligned with the core topics outlined in the title while maintaining a broad and objective perspective on the future of surgical sustainability.
Round 2
Reviewer 3 Report
Comments and Suggestions for Authors
I read with interest the revised version of the manuscript "Advances in Surgery and Sustainability: The Use of AI Systems and Reusable Devices in Laparoscopic Colorectal Surgery." The manuscript has undergone substantial improvement, and I appreciate the authors' efforts in addressing my previous concerns.
I have a few minor suggestions for further refinement:
- In the abstract, there is no need to abbreviate "SDGs" since it is not repeated within the text.
- The section indexing remains somewhat confusing. I recommend avoiding numbered listings and instead maintaining the current division into Chapters 1, 2, and 3.
- While the case series presented may seem supportive of the findings, it holds limited scientific value. I suggest either omitting it or refraining from drawing conclusions based on its results. Additionally, as reusable devices are already widely adopted in laparoscopic practice, providing a clinical rationale for their use seems unnecessary (widely validated thorught real-life practice). Instead, emphasizing their cost-effectiveness through an economic analysis could offer more meaningful insight.
Author Response
Dear Reviewer,
I read with interest the revised version of the manuscript "Advances in Surgery and Sustainability: The Use of AI Systems and Reusable Devices in Laparoscopic Colorectal Surgery." The manuscript has undergone substantial improvement, and I appreciate the authors' efforts in addressing my previous concerns.
I have a few minor suggestions for further refinement:
>> Thank you for your additional insights and constructive feedback. We appreciate your careful review and the effort you have taken to improve the clarity and rigor of our work.
- In the abstract, there is no need to abbreviate "SDGs" since it is not repeated within the text.
>> We agree with your suggestion and have removed the unnecessary abbreviation from the abstract.
- The section indexing remains somewhat confusing. I recommend avoiding numbered listings and instead maintaining the current division into Chapters 1, 2, and 3.
>> We have reorganized Part 3 as the final section (2.3) of Part 2 to improve clarity.
- While the case series presented may seem supportive of the findings, it holds limited scientific value. I suggest either omitting it or refraining from drawing conclusions based on its results. Additionally, as reusable devices are already widely adopted in laparoscopic practice, providing a clinical rationale for their use seems unnecessary (widely validated thorught real-life practice). Instead, emphasizing their cost-effectiveness through an economic analysis could offer more meaningful insight.
>> We agree that the scientific value of the RE-LAC data is limited. We also emphasize that we are not attempting to draw definitive conclusions from this data but rather presenting it as preliminary reference data (line272, line283).
However, there may be differences in interpretation regarding the extent of adoption of reusable energy devices. In our view, "widespread adoption" in a scientific and clinical context means not only that the devices are being used but also that multiple clinical studies have been conducted from various perspectives, leading to the establishment of guidelines for appropriate case selection and verification of device efficacy and safety. From this perspective, we believe that laparoscopic surgery using reusable energy devices has not yet reached this stage. While these devices are indeed used in some medical institutions, comprehensive clinical validation and systematic multi-center studies remain insufficient.
The aim of this review is to explore how to bridge this gap to establish sustainable surgical practices in the future. Specifically, we discuss how reusable energy devices can be more effectively integrated into surgical workflows, how they can be safely implemented on a global scale, and what kind of large-scale clinical studies are necessary to support this process.
We sincerely appreciate your insightful feedback, which has helped us enhance the clarity and focus of our manuscript.